# LOST IN TRANSLATION: CONCEPTUAL BLIND SPOTS IN TEXT-TO-IMAGE DIFFUSION MODELS

## ABSTRACT

Advancements in text-to-image diffusion models have broadened both research and practical applications. However, these models often lead to misalignment issues between text and images. Existing work primarily focuses on problems limited to two objects. We propose a more complex structure, as exemplified by "a tea cup of iced coke." In this instance, an object never previously encountered, a "glass cup," replaces the expected teacup, primarily due to biases in the training dataset. We propose a new classification for such visual-textual misalignment errors, termed Conceptual Blind Spots (CBS). In this study, we employ large language models (LLMs) and diffusion models to thoroughly investigate the diagnosis and remediation of CBS. We develop an automated pipeline that leverages the LLM's proficiency in semantic layering to create a Mixture of Concept Experts (MoCE) framework. To disentangle overlapping concepts, we input them into the models sequentially. Our MoCE is specifically designed to alleviate conceptual ambiguities during the diffusion model's denoising stages. Empirical assessments confirm the effectiveness of our approach, substantially reducing CBS errors and enhancing the robustness and versatility of text-to-image diffusion models.

## 1 INTRODUCTION

In recent years, text-to-image synthesis (Mansimov et al., 2015; Reed et al., 2016; Zhang et al., 2017; Xu et al., 2017; Li et al., 2019; Ramesh et al., 2021; Ding et al., 2021; Wu et al., 2022; Yu et al., 2022; Nichol et al., 2021; Saharia et al., 2022; Rombach et al., 2022; Ramesh et al., 2022; Song et al., 2023) via diffusion models has made remarkable strides, excelling in generating photorealistic images from textual cues (Rombach et al., 2022; Podell et al., 2023). Despite their wide-ranging applications—from digital media to visual arts—a significant limitation exists: the issue of visual-textual misalignment, where certain elements in the input text are overlooked during image synthesis, as our finding in Figure 1 demonstrates. When given inputs like "iced coke" and "tea cup", these models often produce images that only include "iced coke" ignoring the "tea cup" element. More interestingly, the object "glass" that has never been mentioned in the text appears in the image, taking the place of the expected "tea cup". In our study, We term the combination of two main objects (e.g. $\mathcal{A}$:"iced coke", $\mathcal{B}$:"tea cup") a concept pair. Though several studies touch upon the diffusion model's difficulty in generating the correct image while encountering concept pairs, this challenge persists even when sophisticated prompt engineering strategies are employed, indicating a more fundamental issue at play.

In this study, we present an in-depth empirical investigation targeting the conceptual blind spot (CBS) problem, which denotes that when the two objects were presented in a concept pair, either one of them (object $\mathcal{A}$ or $\mathcal{B}$) is encroached upon and consequently disappears in the image. The CBS problem has become a persistent bottleneck that hinders advancements in text-to-image diffusion techniques. More importantly, unlike basic works that only focus on the mutual infringement of $\mathcal{A}$ and $\mathcal{B}$ (Wang et al., 2023; Du et al., 2023; Liu et al., 2022; Li et al., 2023; Chefer et al., 2023),, CBS problem involves a latent object $\mathcal{C}$ that has never been mentioned in the text requirements, such as the "glass" in "a tea cup of iced coke". Its emergence leads to the disappearance of the originally anticipated object $\mathcal{A}$ ("tea cup") in the image, which is due to the strong binding of $\mathcal{C}$ ("glass") with $\mathcal{B}$ ("iced coke") in the training dataset. Our work pivots towards systematically researching the CBS dilemma. Utilizing the reasoning prowess of cutting-edge Large Language Models (LLMs), our

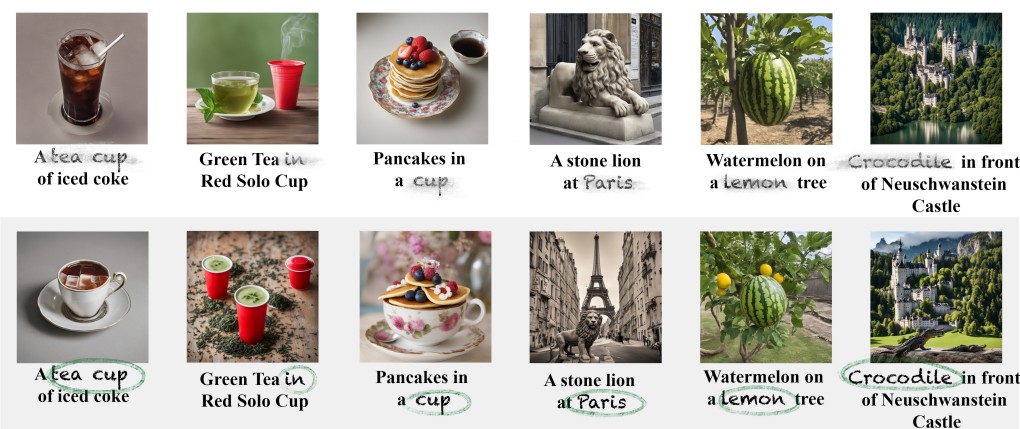

Figure 1: We display the issue of concept misalignment. In the first row of the images, even the most advanced text-to-image models fail to faithfully generate the specified concepts. We have developed a fully automated pipeline to explore this issue and proposed a hotfix to simply fix it.

automated platform accurately identifies and reconciles concept pairings that are prone to generative distortions.

To this end, we adopt Socratic Reasoning (Dong et al., 2023), a process of problem identification, understanding and reasoning using Large Language Models (LLMs). Human experts patiently guide LLMs to discover, understand, and delve deeper into the problem of the issue step by step. We leverage state-of-the-art LLMs to discern and generate an example set of concept pairs of CBS. Upon isolating these concept pairs, we employ a comprehensive evaluation pipeline, employing state-of-the-art diffusion models, to produce high-quality synthesized image samples. Expert human evaluators then carefully assess the resulting images, eliminating concept pairs that display inconsequential discrepancies. We argue that disjunctions between visual and textual components can originate from the complexities of feature space semantics or from biased training sets. To quantitatively assess the misalignments between generated images and specified prompts, we introduce an innovative composite metric that merges proven benchmarks, such as Clipscore (Hessel et al., 2021) and Image-Reward (Xu et al., 2023). This unified metric acts as a reliable gauge for ascertaining the harmony between generated images and their associated textual metadata, offering strong support for our hotfix.

To mitigate the discovered conceptual blind spots, we introduce a more flexible query mechanism, Mixture of Concept Experts (MoCE), to handle intricate constructs like "a tea cup filled with iced coke". Utilizing LLMs, we fine-tune concept sequencing for enhanced image synthesis and optimize dynamic scheduling via heuristic methodologies. Such targeted enhancements not only optimize computational resource allocation but also significantly ameliorate issues related to conceptual discordance. The empirical case studies confirm that our method not only greatly alleviates the CBS challenge but also enhances the application and flexibility of text-to-image diffusion techniques across diverse fields.

We summarize our contributions in the following three aspects:

- We investigate the neglected issue of conceptual blind spots, such as "a tea cup of iced coke", within existing text-to-image diffusion models. Utilizing LLMs for exploration and evaluation, we expose inherent limitations in both architecture and training paradigms.

- We introduce a comprehensive dataset specifically designed to address conceptual blind spots. Employing Socratic-style reasoning, we also outline an automated schema for LLMs to further augment the proposed dataset, stimulating innovation in diverse future directions.

- We propose an adaptive prompting mechanism, diverging from the conventional pre-defined schemes. This adaptability effectively mitigates conceptual discrepancies and visual-textual discordance, thereby enhancing image fidelity and interpretive accuracy.

## 2 RELATED WORK

**Diffusion Models** Diffusion models (Ho et al., 2020) predict noise in noisy images and produce high-quality outputs after training. Having gained considerable attention, they are now considered the state-of-the-art in image generation. These models find applications in various domains, such as label-to-image synthesis (Dhariwal & Nichol, 2021; Rombach et al., 2022), text-to-image (*ti*) generation (Ramesh et al., 2022; Rombach et al., 2022; Podell et al., 2023), image editing (Couairon et al., 2022; Meng et al., 2021; Kawar et al., 2023), and video generation (Ho et al., 2022). Specifically, the synthesis of *ti* images is of practical importance. Open-source and commercial solutions like Stable Diffusion (Rombach et al., 2022; Podell et al., 2023) and Mid-journey (*mj*) (Midjourney, 2023) have achieved success, bolstered by advancements in neural network architectures (Ronneberger et al., 2015; Vaswani et al., 2017) and a large dataset of image-text pairs (Sharma et al., 2018; Schuhmann et al., 2022). Nevertheless, the question of whether these models truly innovate or merely generate combinations encountered in their training data remains unresolved. This study investigates this issue by examining text-to-image diffusion models in the context of unconventional concept pairings.

**Misalignment Problems** While prominent generative models frequently produce high-quality, realistic images, they struggle with certain concept combinations. Such models usually mimic combinations seen in training data. Prior work has emphasized spatial conflicts where multiple entities coexist in close proximity (Wang et al., 2023; Du et al., 2023; Liu et al., 2022; Li et al., 2023; Chefer et al., 2023). Distinct from these investigations, our focus lies on conceptual blind spots, illustrated by phrases such as "a tea cup of iced coke." Through rigorous experimentation, we investigate this challenge and introduce a benchmark alongside a baseline solution.

## 3 BENCHMARK: COLLECTING DATA ON CONCEPTUAL BLIND SPOTS (CBS)

In this section, we describe the construction of our dataset, CBS, focusing on the conceptual gaps in image generation for two distinct concepts, $\mathcal{A}$ and $\mathcal{B}$, and the influence of the latent concept $\mathcal{C}$. Collecting such data requires an extensive range of knowledge, which is typically unrealistic for the human brain. Therefore, we have constructed Socratic Reasoning process through the collaboration of generative models and human experts. Specifically, drawing inspiration from Dong et al. (2023), a Socratic Reasoning system is implemented using GPT-3.5, text-to-image models, and human researchers to investigate issues of visual-textual misalignment. The framework comprises:

- **GPT-3.5** serves as an expert rich in information but lacking comprehensive wisdom. It elaborates on issues when prompted, leveraging its intrinsic knowledge and insights.

- **Text-to-image models** (T2I models) functions as an experienced artist with limited creative abilities. Its main role is to generate images based on understood prompts. We utilize both Midjourney and SDXL-1.0, confirming the consistent occurrence of the problem.

- **Human researchers** act as discerning advisors, overseeing the outputs of GPT-3.5 and T2I models. They provide guidance to GPT-3.5 and extract insights to deepen their understanding of the misalignment problem, thereby setting the stage for future refinements.

To assess the accuracy of the generated images, human researchers introduce a metric system. Specifically, T2I models generate 20 images per concept pair for evaluation. In this system, zero correct images correspond to Level 5, $1 \sim 5$ correct images to Level 4, $6 \sim 10$ to Level 3, $11 \sim 15$ to Level 2, and $16 \sim 20$ to Level 1.

The Socratic Reasoning process begins with human researchers identifying error-prone concept pairs. Subsequently, GPT-3.5 generates additional concept pairs. Human researchers then prompt GPT-3.5 to produce new patterns, culminating in a closed-loop Socratic Reasoning system after integrating various patterns. These stages are distributed across six rounds. The idea of the construction of these six rounds is composed using a divide-and-conquer strategy. In the process, we leverage human experts' creativity and induction to initialize the few-shot data and categorize the patterns and the extensive knowledge possessed by GPT to expand our dataset. Here, human experts are involved only in the initial two stages. Afterwards, GPT operates like a snowball effect, continually expanding the scope of data collection.

Table 1: Categories and patterns summarized manually by human researchers.

| Category | Pattern |
| --- | --- |
| I: Dominant Association | Local cuisine and non-native location 
 Endangered species and historical sites 
 Animal and incorrect external covering 
 Beverage and erroneous container |
| II: Absence of Dominant Association | Inappropriate tablewares and food 
 Visual nouns of similar shapes 
 Animal nouns with similar appearances 
 Mismatched outfits |

## 3.1 ROUND 1 - IDENTIFYING 259 CONCEPT PAIRS FROM TEXT-TO-IMAGE DATASETS

Initially, 259 concept pairs are identified through the collaborative efforts of human researchers, supported by GPT-3.5. These pairs undergo rigorous manual classification. Specific examples, such as "a tea cup of iced coke", inform our investigation into root causes. Prior to expanding the dataset with analogous concept pairs for comprehensive analysis, it is crucial to understand the fundamental patterns exhibited in typical concept pairs. To this end, human researchers examine various visual scenes in well-known text-to-image datasets such as Laion2B-en[1] and MJ User Prompts & Generated Images Dataset[2] to identify concept pairs susceptible to similar errors.

Human researchers come up with 50 concept pairs as initial seeds for potential erroneous outputs. These seed pairs are then segregated by human researchers into two primary categories and eight specific patterns. This classification approach aims for conceptual clarity and addresses the pervasive issue of concurrency in such models. As delineated in Table 1, the updated text offers a comprehensive categorization criterion, explores the challenges associated with each category, and establishes a systematic framework for subsequent evaluations:

**Category I: Dominant Association** In this category, one concept (e.g., $\mathcal{B}$) in the model's training dataset is strongly bound to a latent concept $\mathcal{C}$, resulting in the influence on another expected object $\mathcal{A}$ in the image. For instance, the $\mathcal{A}$ ("iced coke") contain the underlying concept of $\mathcal{C}$ ("glass cup"), enconpassing the influence of $\mathcal{B}$ ("tea cup")

**Category II: Absence of Dominant Association** This category includes instances where no apparent hierarchical or dominant relationship exists between the entities. The two entities often exist only in mutual encroachment or integration.

Based on this classification, we can thus aim our overall goal of the dataset to collect CBS samples with the inclusion of a latent concept $\mathcal{C}$ mentioned above, which is a focus not emphasized in previous work. Besides that, a method is devised to collect a substantial number of incorrect concept pairs. Human researchers employ the GPT-3.5 API to generate various concept pairs that align with each pattern in Table 1. Using this approach, GPT-3.5 generates 270 concept pairs, of which 209 result in image synthesis following prompts formatted as "$\mathcal{A}$, $\mathcal{B}$". Each synthesized image contains errors. This initial observation validates our hypotheses concerning error patterns depicted in Table 1. Including the 50 concept pairs manually extracted, a total of 259 error-prone concept pairs are identified in this round.

## 3.2 ROUND 2 - RIGOROUS VERIFICATION ON DISCOVERED 259 CONCEPT PAIRS

In this round, human researchers identify 159 Level 5 concept pairs from the initial set of 259, following rigorous verification. In previous iterations of T2I models, detailed descriptions are essential for generating high-quality images, implying that granularity enhances image accuracy. Simple comma concatenations between concepts may be insufficient. Human researchers recommend including comprehensive details for each concept pair to enable a more robust evaluation. Utilizing

---

[1]https://huggingface.co/datasets/laion/laion2B-en
[2]https://huggingface.co/datasets/succinctly/midjourney-prompts

LLM at this stage enhances contextual understanding, aligning it more closely with human cognition.

Specifically, for each concept pair, human researchers utilize GPT-3.5 to generate the top five scene description sentences, usually incorporating verbs. Consequently, $259 \times 5 = 1295$ sentence-based text prompts are fed into T2I models, which in turn produces $1295 \times 4 = 5180$ images. After rigorous screening, 159 concept pairs attain a Level 5 rating, representing the pinnacle of quality.

### 3.3 ROUND 3 - IDENTIFYING 240 ADDITIONAL CONCEPT PAIRS

A total of 159 concept pairs serve as examples, with GPT-3.5 prompted to generate an additional 240 pairs. Following stringent validation, 113 concept pairs attain a Level 5 rating. The insights gained from GPT-3.5 contribute significantly to exploration, offering valuable examples that guide GPT-3.5 in producing superior outputs. Specifically, human researchers employ GPT-3.5 to augment the dataset comprising error-prone concept pairs. Top-performing pairs from Round 2 serve as positive instances for in-context learning, while the earliest eliminated pairs function as negative instances. Consequently, GPT-3.5 generates 30 unique concept pairs for each of the eight pre-defined patterns, resulting in 240 new pairs. Human researchers then replicate the validation procedures from Rounds 1 and 2. The culmination of Round 3 in the Socratic Reasoning framework is the identification of 113 concept pairs rated at Level 5. Note that GPT-3.5 may still generate duplicates when asked to create additional concept pairs, despite instructions to avoid them.

### 3.4 ROUND 4 - MINING 9 NEW PATTERNS FROM 8 KNOWN PATTERNS

Based on the existing two categories and eight finely-sorted patterns, human researchers direct GPT-3.5 to generate nine additional patterns. The recurring emergence of redundant concepts may signify that the boundaries of GPT-3.5's knowledge within the current pattern have been reached. Leveraging Socratic Reasoning, when encountering limitations in one dimension, the natural inclination is to explore new dimensions, thereby discovering novel patterns. Specifically, human researchers pose a challenge to GPT-3.5, prompting it to independently derive new patterns, using the existing eight patterns as a reference. Given that Category I has a wider scope than Category II, human researchers instruct GPT-3.5 to develop six new patterns for Category I and three for Category II. Remarkably, GPT-3.5 comprehends our objectives and furnishes new patterns, complete with an accompanying explanation and an illustrative concept pair. For further details, please refer to the Appendix.

### 3.5 ROUND 5 - VERIFYING 9 NEW PATTERNS DISCOVERED BY LLMS

With the revised methodology, nine new patterns attain a Level 5 ratio exceeding 10 out of 15 concept pairs. The autonomy of GPT-3.5's work remains a priority. Thus, human researchers abstain from supplying concept pairs as guidance for emerging patterns. Instead, they replicate the Round 3 procedure, withholding examples to encourage GPT-3.5 to autonomously generate 15 concept pairs. Using the "Jewelry and Improper Storage" pattern in Category I as a case study, only one out of the 15 concept pairs achieves a Level 5 rating. It is acknowledged that in the absence of visually validated examples, GPT-3.5 might face challenges in making precise inferences. Is it feasible to steer GPT-3.5 from a linguistic space standpoint?

Our objective is for human researchers to act as facilitators rather than prescriptive overseers. To that end, a transition is proposed to focus on underlying concepts within the linguistic sphere instead of specific concept pairs. In this context, human researchers are required only to select relevant conflicting concepts from two concept sets, leveraging their experience in text-image interactions, before subjecting them to verification. This approach circumvents both aimless exploration and undue guidance. Based on their understanding, human researchers select 15 combinations from two sets of 20 concepts, such as "Ruby Earrings, Bird Nest", "Pearl Necklace, Fish Tank", and "Silver Anklet, Toilet Bowl". They then execute the Round 2 procedure, generating 75 text prompts in the form of sentences for T2I models. After verifying the $75 \times 4 = 300$ generated images, 92 out of 135 concept pairs receive a Level 5 rating, significantly surpassing the proportion achieved by GPT-3.5 without example-based guidance.

### 3.6 ROUND 6 - BLENDING PATTERNS FOR NEXT CYCLE OF DATA COLLECTION

All 17 patterns serve as the basis for synthesizing blended concept pairs, generating innovative pattern combinations. In each cycle of data collection, we guide the development of a new conceptual space using GPT-3.5's capabilities. Specifically, human researchers direct GPT-3.5 to integrate each of the nine emergent patterns with an existing set of eight. For example, patterns labeled "Beverage and Incorrect Container" and "Jewelry and Inadequate Storage" are merged to create new frameworks: "Beverage and Jewelry Storage" and "Jewelry and Beverage Container". Following Round 5 protocols, we find that each newly formed pattern achieves a Level 5 ratio in at least 10 of the 15 evaluated concept pairs, confirming GPT's effectiveness in merging orthogonal patterns into a more sophisticated and resilient structure.

In summary, faced with an intentionally introduced anomalous pattern, LLM like GPT-3.5 undergoes thorough Socratic questioning led by human researchers as examiners. This repetitive process encompasses the generation of as many valid concept pairs as possible within the existing pattern, the extraction of crucial insights, the formation of an enhanced pattern, and the subsequent amalgamation of insights from both patterns to expand the data's application scope. The system allows for unlimited iterations through this exacting workflow. Regarding the limits of LLM's capabilities, this remains an unresolved issue, answerable only through future research.

## 4 METHOD: MIXTURE OF CONCEPT EXPERTS (MoCE)

**Background** Diffusion Models (Sohl-Dickstein et al., 2015; Ho et al., 2020) modify data by progressively adding noise to an initial state $x_0 \sim q(x_0)$. This process is formulated as a Markov chain $q(x_{1:T}|x_0) = \prod_{t=1}^{T} q(x_t|x_{t-1}), q(x_t|x_{t-1}) = \mathcal{N}(x_t|\sqrt{\alpha_t}x_{t-1}, \beta_t I)$, where $I$ denotes the input image, and $\alpha_t = 1 - \beta_t$ characterizes the noise schedule.

Text-to-image generation strives to understand the conditional distribution $p(\text{Image}|\text{Text})$. Given this framework, paired data $(x_0, y_0) \sim q(x_0, y_0)$ exists, and the objective is to represent the conditional data distribution $q(x_0|y_0)$. The Gaussian model for the reverse transition, dependent on $y_0$, is represented as $p(x_{t-1}|x_t, y_0) = \mathcal{N}(x_{t-1}|\mu_t(x_t, y_0), \sigma_t^2 I)$.

**MoCE** In a text-to-image task, multiple entity concepts are often provided to the generation model simultaneously. However, these concepts can potentially conflict, leading to only stronger entities prevailing in the final output. Recognizing this, we aim to devise a structured method ensuring the representation of each entity. Motivated by dynamic models (Han et al., 2021), we integrate the Mixture of Experts (MoE) framework (Wang et al., 2020), offering varied concepts as cues for diffusion models at distinct time intervals. Importantly, GPT-3.5 will autonomously decide a logical drawing order for each text cue. We then alleviate alignment issues by introducing "Mixture of Concept Experts" (MoCE). We base MoCE on SDXL-1.0 (Podell et al., 2023), one of the foremost reliable open-source diffusion models. The system includes the subsequent essential components:

**Sequential Concept Introduction** Drawing inspiration from human artistic processes, entities are introduced to diffusion models sequentially rather than simultaneously to prevent entanglement. GPT-3.5 is employed to ascertain the most logical sequence for introducing entity concepts to diffusion models based on its comprehension of human behavior. We provide additional interaction details with GPT-3.5 in the Appendix; please refer to it for further information.

**Automated Feedback Mechanism** In the denoising process of diffusion models across time steps, $t_T, t_{T-1}, ..., t_1, t_0$, the process is bifurcated into two distinct phases. In the initial phase, we furnish the model with prompts delineating the concepts to be introduced initially. In the succeeding phase, we supply the model with the complete text prompts. This approach motivates us to formulate a strategy for autonomously determining the optimal allocation of time steps.

Clipscore, $\mathcal{S}_c$, is used to assess the fidelity of the generated images, $M$, to the desired concepts, $\mathcal{A}$ and $\mathcal{B}$:

$$\mathcal{S}_c(M, \mathcal{A}) = ClipScore(M, \mathcal{A}) \tag{1}$$

Considering the issue of misalignment, employing Clipscore naively may not yield effective results. Therefore, we have implemented two improvements to address this concern. First, we encourage

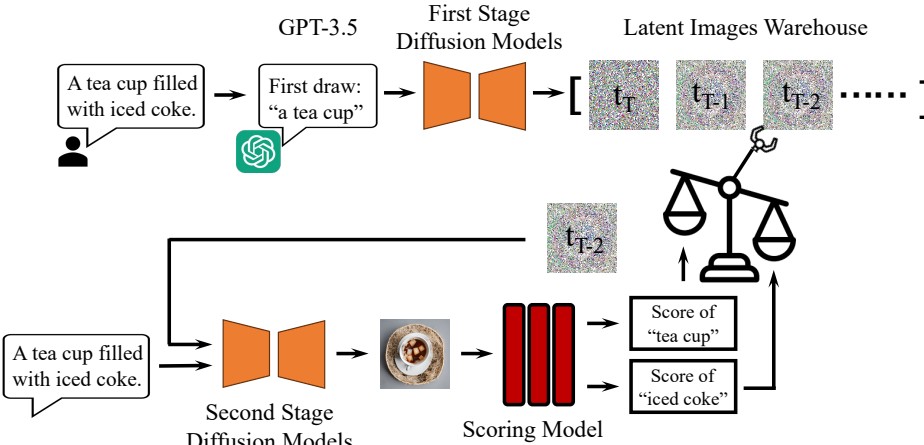

Figure 2: Overview of our method, MoCE. When provided with a text prompt, GPT-3.5 determines which concept should be drawn first. In the first stage, SDXL-1.0 generates latent images at each step, storing them in a warehouse. The second stage model takes one of these images and performs the second stage of denoising. Scores of two concepts are then used to select a better image from the warehouse. This selection process is aided by a binary search algorithm.

GPT-3.5 to generate descriptions for each concept and calculate the Clipscore between images and both the concept itself and its corresponding description. We consider the higher of these two scores as the score for images and the concept:

$$\mathcal{S}(M, \mathcal{A}) = Max(\mathcal{S}_c(M, \mathcal{A}), \mathcal{S}_c(M, \mathcal{A}_{Description})) \tag{2}$$

Second, we have devised a new metric, denoted as $\mathcal{D}$:

$$\mathcal{D} = \mathcal{S}(M, \mathcal{A}) - \mathcal{S}(M, \mathcal{B}) \tag{3}$$

$\mathcal{D}$ denotes the difference in the scores of an image between two concepts. The greater the absolute value of this difference, the higher the probability that one of the concepts is not clearly represented.

When allocating a larger number of time steps to the first stage, the features of $\mathcal{A}$ in the produced images become more dominant, possibly overshadowing the features of $\mathcal{B}$. In contrast, if the allocation favors the second stage, the features of $\mathcal{A}$ could be overshadowed by $\mathcal{B}$. Therefore, there is a linear relationship between the time steps allocated in the first phase and the image quality, making the binary search method highly suitable for this scenario. Specifically, we initialize the boundaries for the left and right time steps, represented as $\mathcal{L}$ and $\mathcal{R}$. Subsequently, we use $\mathcal{M} = \frac{\mathcal{L}+\mathcal{R}}{2}$ as the dividing point between stage 1 and stage 2 for image production. If the score for $\mathcal{A}$ exceeds that of $\mathcal{B}$ at this juncture, $\mathcal{M}$ is set as the new $\mathcal{R}$, signifying a shift in the binary search's direction. The algorithm then repeats this procedure until either both scores attain a predetermined threshold or the scores converge closely.

To enhance the system's performance, we dynamically adjust the number of sampling steps, as depicted in Figure 2. We also prepare intermediate images in the first stage for direct selection in the second stage, thereby limiting iterations to the second stage and saving approximately one-third of the computational time without additional memory usage.

By integrating these components, our methodology presents a robust solution for addressing entity misalignment in text-to-image diffusion models. It combines intuitive reasoning, automatic fine-tuning, and efficiency optimizations to produce more precise and contextually apt image outputs.

## 5 EXPERIMENTS

We conduct extensive experiments centered around our MoCE, revealing its capability to alleviate concepts absent in the original generations of text-to-image diffusion models.

## 5.1 SETUP

**Datasets**  As introduced in Section 2, Category II misalignment issues have been extensively discussed and resolved. Conversely, our approach specifically targets misalignment within Category I. After the initial three rounds of the Socratic method, we acquired a total of 272 concept pairs for Level 5. Of these, we focus on 173 pairs for Level 5 in Category I. As established in Section 3, Level 5 indicates that none of the 20 generated images in a pair is correct. In this section, we conduct experiments on these Category I concept pairs using our proposed method. We select the shortest text prompt from each concept pair's five sentences to highlight our method's effectiveness.

**The Model**  Our remediation primarily targets SDXL-1.0 (Podell et al., 2023) due to its white-box nature and widespread use. We omit Midjourney from consideration as its black-box architecture prohibits internal state modification for dynamic prompting. Tests indicate comparable performance, with fewer than $10\%$ of samples yielding divergent outcomes when deploying both models to address misalignment issues. Consequently, concept pairs derived from Midjourney remain applicable to SDXL-1.0. Regarding computational resources, our experiments utilize a single NVIDIA A100 GPU with 80GB memory for image generation via SDXL-1.0.

Furthermore, we introduce 2 improved methods as additional baselines: AAE (Chefer et al., 2023) and DALLE-3[3]. AAE aims to mitigate examples in Category II from the perspective of the attention map layer, while DALLE-3 attempts to alleviate misalignment issues through fine-grained annotation of the training data.

**Evaluation Metrics**  Considering the instability of score evaluation, as well as the use of Clipscore and Image-Reward in the Method, we primarily utilize human evaluation in our experiments. We engage human experts for a more impartial evaluation. Additionally, the results of score evaluation are included in the Appendix.

## 5.2 RESULTS

**Human Evaluation**  Human experts assess the images generated by our MoCE using concept pairs from Level 5. They then re-evaluate these pairs based on the criteria discussed in Section 3. We report the counts of concept pairs at each level after undergoing improvement by our MoCE and compare them to the baseline results in Tabel 2.

Table 2: Human Evaluation for our MoCE. Human experts re-rate the concept pairs based on the images generated by our MoCE.

| Method | New Level | | | | |
| --- | --- | --- | --- | --- | --- |
| | Level 1 ($\uparrow$) | Level 2 ($\uparrow$) | Level 3 ($\uparrow$) | Level 4 ($\uparrow$) | Level 5 ($\downarrow$) |
| SDXL-1.0 Baseline | 0 | 0 | 0 | 0 | 173 |
| AAE Baseline | 0 | 0 | 6 | 10 | 157 |
| DALLE-3 Baseline | **14** | 24 | 31 | 23 | 81 |
| MoCE (ours) | 11 | **25** | **40** | **71** | **26** |

In the original Level 5 concept pairs, the baseline model fails to produce any correct image. However, following the enhancement made by our MoCE, over half of the concept pairs are now correctly generated, and there are even several concept pairs rated as Level 1. Furthermore, even when sophisticated engineering strategies, like AAE, are used, which may be effective for traditional Category II data, there is little improvement for Level 5 concept pairs when applied to our Category I (CBS) data. Additionally, DALLE-3, with its expensive and fine-tuned data annotations, indeed helps with the CBS problem, achieving improvements for Level 5 concept pairs comparable to MoCE. However, it is important to note that training DALLE-3 requires additional data preprocessing, which can be a costly process, suggesting that our MoCE produces a greater number of correct images in an economical and effective manner.

---

[3] OpenAI. Dall·e 3 system card. 2023

## 5.3 ANALYSIS

Clipscore: 0.660  Clipscore: 0.659  Clipscore: 0.679  Clipscore: 0.634  Clipscore: 0.663  Clipscore: 0.670
Image-Reward: 0.061  Image-Reward: 0.370  Image-Reward: 0.029  Image-Reward: -0.383  Image-Reward: -0.271  Image-Reward: 0.397

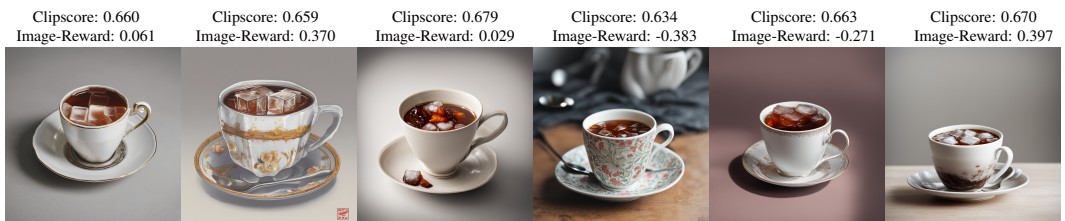

(a) Generated by our method, MoCE.

Clipscore: 0.738  Clipscore: 0.667  Clipscore: 0.732  Clipscore: 0.738  Clipscore: 0.754  Clipscore: 0.764
Image-Reward: 0.739  Image-Reward: 1.153  Image-Reward: 1.460  Image-Reward: 1.305  Image-Reward: 1.531  Image-Reward: 1.474

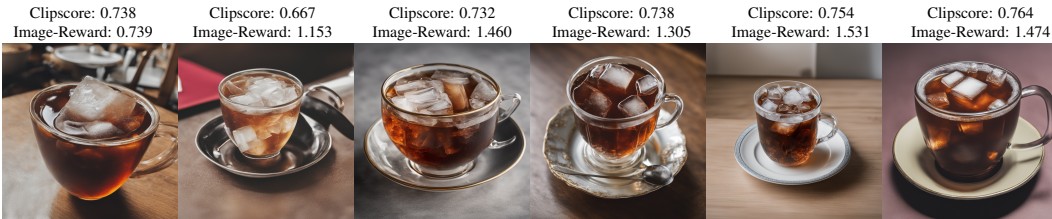

(b) Generated by baseline model, SDXL-1.0.

Figure 3: Visualization of images of *a tea cup of iced coke* generated by both our MoCE and baseline model. We also report Clipscore (↑) and Image-Reward Score (↑) between images and the concept, *iced coke*, to demonstrate the minor pitfalls of the existing evaluation metrics.

For the most representative and challenging example, "a tea cup of iced coke", we present visual restoration results in Figure 3. Further restoration visualizations are available in the Appendix; interested readers are referred to this section for additional details. Both Clipscore and Image-Reward effectively adjust the time step in our MoCE model. Unfortunately, existing evaluation metrics may occasionally prove insufficient. Figure 3 displays both Clipscore and Image-Reward scores between images and the concept "iced coke". We meticulously chose transparent glasses generated by baseline models, which resemble tea cups, to analyze the error in the scoring mechanism. The occurrence of "iced coke" in both sub-figures within Figure 3 is evident to human experts. However, both Clipscore and Image-Reward scores are markedly lower for images of "a tea cup of iced coke" attributed to the material of the cup. This highlights the limitations of current evaluation metrics in addressing misalignment issues and emphasizes the necessity for developing new metrics based on existing methods.

## 6 CONCLUSION

In this paper, we introduce a novel framework for addressing the text-to-image synthesis challenge of conceptual blind spots (CBS). Leveraging Large Language Models (LLMs), our framework excels in identifying problematic concept pairings commonly found in generative tasks. Our key innovation, the Mixture of Concept Experts (MoCE), enhances the flexibility and accuracy of image generation, effectively handling complex constructs like "a tea cup of iced coke". We validate our approach using a robust evaluation pipeline that incorporates state-of-the-art diffusion models and a composite metric, integrating established benchmarks such as Clipscore and Image-Reward. Empirical results, corroborated by expert human evaluations, confirm the efficacy of our contributions in mitigating conceptual blind spots.

**Reproducibility Statement**  To promote the reproducibility of our research, we have undertaken the following measures:

- Interaction protocols with LLMs are delineated in Section 3 and the Appendix.
- A comprehensive exposition of our methodology is provided in Section 4.
- The Supplementary Material contains both the source code and the curated concept pairs.

These resources facilitate straightforward replication of our study.

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

## APPENDIX

In the appendix, we initially elucidate a comprehensive example that employs Socratic Reasoning to harness Large Language Models (LLM) for the advancement of scientific discovery. Subsequently, we outline a methodology for directing ChatGPT in the generation of incremental instructions aimed at the seamless introduction of sequential concepts. Then, we offer a series of visual aids to shed light on the pervasive issue of dominant concept overlay, spanning Levels 1 through 5. Lastly, we offer a presentation of concept pairs rated as Level 5 in our dataset, the sentences describing them, descriptions of concepts and the input sequence provided by GPT-3.5.

## A    INTERACTION DETAILS IN SOCRATIC REASONING

In this section, we describe the detailed process of interaction between human researchers and GPT-3.5 within the Socratic Reasoning framework. Utilizing block diagrams for clarity, we model the engagement between human experts and GPT-3.5, based on the methodology set forth by Dong et al. (2023). To more clearly illustrate the variation in data volume within Socratic Reasoning, we present several diagrams, in Figure 4. In these diagrams, pairs of spheres along the same axis denote concept pairs associated with a particular pattern, distinguishable by their color.

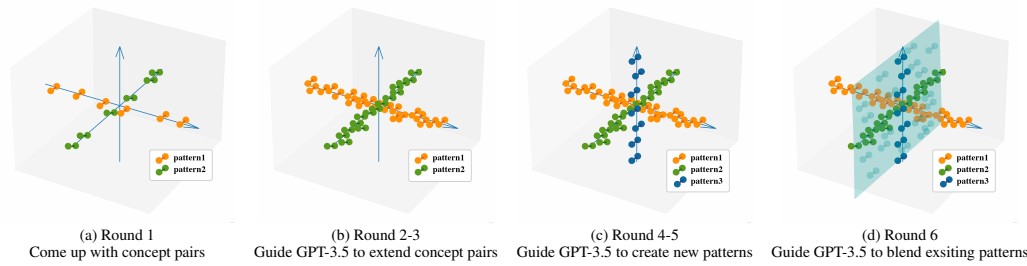

(a) Round 1
Come up with concept pairs

(b) Round 2-3
Guide GPT-3.5 to extend concept pairs

(c) Round 4-5
Guide GPT-3.5 to create new patterns

(d) Round 6
Guide GPT-3.5 to blend exsiting patterns

Figure 4: Schematic diagrams of different rounds of Socratic Reasoning. Pairs of spheres along the same axis denote concept pairs associated with a particular pattern A complete Socratic Reasoning contains coming up with concept pairs (Figure a), guiding GPT-3.5 to extend concept pairs (Figure b), guiding GPT-3.5 to create new patterns (Figure c) and guiding GPT-3.5 to blend exsiting patterns (Figure d).

**Round 1: Extracting Concept Pairs from Text-to-Image Datasets**    After developing an initial set of 50 concept pairs and categorizing them into eight unique patterns, human researchers direct GPT-3.5 to generate additional concept pairs. We specify both the researcher-provided prompt and the subsequent GPT-3.5 response. Midjourney text-to-image generation, in conjunction with human researcher validation, confirms a total of 270 concept pairs. Among these, 209 pairs satisfy the validity criteria, resulting in a comprehensive set of 259 error-sensitive concept pairs at the end of the first Socratic round.

> **1**
>
> **Prompt:** I want you to act as an AI creative assistant helping users generate pairs of nouns on cue: pairs of Beverage and erroneous container (e.g. iced coke, tea cup). Each pair occupies one line, in 1 ~ 3 words, separated by commas.
>
> **Answer:** Certainly, I can help you generate pairs of nouns as requested. Here are some pairs of beverages and wrong containers:
>
> 1. Coffee, Champagne Flute
> 2. Orange Juice, Coffee Mug
> 3. Hot Chocolate, Martini Glass
> 4. Iced Tea, Shot Glass
> 5. Lemonade, Beer Stein
> ......
>
> These pairs should provide some creative and humorous combinations of beverages and the erroneous containers for your use. If you need more pairs or have any specific preferences, feel free to let me know!

**Round 2 - Rigorous Verification on the 259 concept pairs** Ongoing advances in Midjourney development indicate that integrating larger and more diverse datasets improves the quality of Midjourney's image generation. To rigorously test this hypothesis, researchers utilize GPT-3.5 to generate five context-rich sentences for each of the 259 concept pairs studied. Consequently, a collection of 1,295 text prompts, structured as complete sentences, is formulated. These prompts guide Midjourney in generating a corpus of 5,180 unique images. Following meticulous human evaluation, the second version of the Socratic Reasoning framework reveals that of the concept pairs examined, 159 meet Level 5 criteria, 75 meet Level 4, 15 meet Level 3, and 7 meet Level 2. Additionally, a smaller subset of 3 pairs is classified at Level 1.

> **2**
>
> **Prompt:** I want you to act as an AI creative assistant helping users generate images using Midjourney. Your job is to generate a prompt for text-to-image generation model. You are good at text-image concept grounding, visual concept understanding, text-to-image diffusion models' prompt engineering. Don't be verbose in your answers, only provide the proposed prompt itself and avoid including explanations. I've provided you with a pair of visual concepts, and I want you to provide me with detailed prompts to feed into Midjourney that one given concept would show up with the other one in the corresponding generated image. It will be your job to come up with five prompts that closely integrate two concepts visually, yet each remains a distinct entity. Remember there is NO extra object or human in the prompt, NOR any explanatory sentences, according to the user's specifications.
> Now, the given two visual concepts are: Iced coke, tea cup
>
> **Answer:**
> 1. Ice coke bubbling in a tea cup.
> 2. A tea cup of ice coke.
> 3. Ice coke poured into a tea cup.
> 4. Iced coke in a tea cup.
> 5. Tea cups filled with iced coke, cold fogging up.

**Round 3 - Identifying 240 Additional Concept Pairs** The efficacy of GPT-based models for information extraction is well-established. Given an adequate dataset of high-quality concept pairs (Level 5), these models can be fine-tuned to autonomously generate additional concept pairs following predetermined patterns. The model generates 30 unique concept pairs for each of the eight patterns, thus avoiding concept duplication. However, scalability poses a challenge; despite constraints, increased demand may result in concept repetition. This approach yields $30 \times 8 = 240$ distinct concept pairs in total. In subsequent iterations, as shown in Round 3 of the Socratic method, the distribution varies: 113 pairs at Level 5, 47 pairs at Level 4, four pairs at Level 3, one pair at Level 2, and none at Level 1.

3

**Prompt:** I want you to act as a concept pair generator for text-to-image generation model, midjourney. You are good at text-image concept grounding, visual concept understanding, text-to-image diffusion models' prompt engineering. Don't be verbose in your answers, but do provide details and examples where it might help the explanation. I've provided you with a bunch of existing positive and negative noun-concept pairs. The existing positive pairs comprise a common beverage and an unconventional common container that is not typically used to hold or serve that beverage, making them positive pairs. Specifically, given noun-concept pair like "iced coke" and "tea cup", they are both common in the real world. However, the resultant text-to-image synthesis unexpectedly refers to "a glass cup containing iced coke", with the concept of "tea cup" ignored. It will be your job to come up with 30 such new positive noun-concept pairs following the given pairs. You need to remember that nouns cannot be repeated between different concept pairs.

[positive] Coffee, Champagne Flute
[positive] Orange Juice, Coffee Mug
[positive] Iced Tea, Shot Glass
[positive] Milkshake, Whiskey Tumbler
[positive] Coca-Cola, Water Bottle
[positive] Green Tea, Red Solo Cup
[positive] Hot Coffee, Cocktail Shaker
[positive] Iced Coffee, Shot Glass
[positive] Pina Colada, Beer Stein
[positive] Iced Latte, Soda Can
[positive] Espresso, Water Bottle
[positive] Hot Tea, Coffee Mug
[positive] Hot Apple Cider, Cocktail Shaker
[positive] Hot Chocolate, Red Solo Cup
[positive] Pumpkin Spice Latte, Water Bottle
[positive] Mojito, Highball glass

[negative] Smoothie, Tea cup
[negative] Iced Cola, Glass
[negative] Orange Juice, Juice Glass
[negative] Hot Coffee, Coffee Mug
[negative] Beer, Beer Mug
[negative] Lemonade, Pitcher
[negative] Water, Water Bottle
[negative] Wine, Wine Glass
[negative] Iced Mocha, Soda Can
[negative] Cold Brew, Wine Glass
[negative] Mojito, Pint Glass
[negative] Apple Cider, Champagne Flute
[negative] Iced Almond Milk Latte, Pint Glass

**Answer:** Sure! Based on the examples you provided, I will generate 30 new positive noun-concept pairs (beverage, uncommon vessel). I will ensure that the nouns are not repeated. Here are the pairs:

1. [positive] Ginger Ale, Martini Glass
2. [positive] Root Beer, Teapot
3. [positive] Soda, Whiskey Flask
4. [positive] Chocolate Milk, Wine Carafe
5. [positive] Matcha Latte, Beer Boot
6. [positive] Bloody Mary, Coffee Carafe
7. [positive] Seltzer, Whiskey Decanter
8. [positive] Lemon Water, Margarita Glass
9. [positive] Sprite, Tea cup
10. [positive] Lassi, Martini Shaker
11. [positive] Fruit Punch, Whiskey Flask
12. [positive] White Wine, Shot Glass
13. [positive] Champagne, Milk Jug
14. [positive] Red Wine, Milkshake Glass
15. [positive] Diet Coke, Teapot
16. [positive] Pepsi, Champagne Bucket
17. [positive] Mountain Dew, Wine Decanter
18. [positive] Black Tea, Margarita Glass
19. [positive] Mint Julep, Coffee Carafe
20. [positive] Rum, Tea Kettle
21. [positive] Whiskey, Juice Box
22. [positive] Vodka, Soup Bowl
23. [positive] Gin and Tonic, Coffee Thermos
24. [positive] Sprite, Wine Bucket
25. [positive] Protein Shake, Wine Carafe
26. [positive] Kombucha, Martini Glass
27. [positive] Earl Grey Tea, Champagne Bucket
28. [positive] Cappuccino, Margarita Glass
29. [positive] Fanta, Coffee Pot
30. [positive] Tonic Water, Teapot

Please note that these pairs are meant to be unexpected and not typical combinations, which is the point of the exercise.

**Round 4 - Mining 9 New Patterns** The proliferation of repetitive themes has prompted researchers to investigate innovative approaches. Engaging with machine learning models such as GPT-3.5, scholars aim to automate the discovery of new patterns. GPT excels at identifying the underlying features of these novel frameworks, extending and refining pre-existing methodologies. It not only detects but also rigorously validates emerging patterns within the realm of linguistic constructs, often corroborating its findings through pertinent examples.

> **4**
>
> **Prompt:** I want you to act as a bot summarizing the pattern rule and generating more patterns. You are good at text-image concept grounding, visual concept understanding, text-to-image diffusion models' prompt engineering. Don't be verbose in your answer, but do provide details and examples that might help the explanation.
>
> I have provided you with a series of existing positive and negative patterns, where "pattern" refers to a generalization of certain noun pairs, A and B. Where A and B are very common concepts, such as coke and tea cup. Moreover, these noun pairs are closely related to visual imagery, rather than being two independent objects, such as the wrapper and the wrapped item, background and foreground, etc.
>
> In the provided positive and negative patterns, the noun pairs in the positive pattern are combinations that almost never appear in the text-to-image training dataset, such as beverage and erroneous container (e.g., iced cola and tea cup) or local cuisine and non-native location (e.g., sushi and New York City). Specifically, for a positive noun pair "A" and "B", there will usually be a hidden concept "C". "A" always appears with "C" in the text-to-image training dataset, "B" and "C" belong to the same category or are of the same kind but their visual effect is very different. Therefore, because of the strong association between "A" and "C," it leads to "B" being obscured or occupied in the generated image with prompts "A" and "B". For example, in the positive pattern "Beverage and erroneous container", "A" (iced coke) and "B" (tea cup) are a good example pair. Because "A" (iced coke) is strongly bound to the latent concept "C" (glass), telling the model to generate "a tea cup of iced coke" will generate a glass of iced coke instead of a tea cup of iced coke.
>
> Conversely, the noun pairs in the negative pattern are combinations that frequently appear in the text-to-image training dataset, such as animal and their corresponding external covering or jackets and their appropriate inner garments. Your task is to come up with some new positive patterns based on the given positive and negative patterns:
>
> [positive] Beverage and erroneous container
> [positive] Animal and incorrect external covering    [negative] Food and its appropriate container
> [positive] Mismatched outfits    [negative] Furniture and its corresponding setting
> [positive] Endangered species and historical sites    [negative] Endangered animals and their habitats.
> [positive] Local cuisine and non-native location
>
> **Answer:** Certainly, here are some new positive patterns based on the provided guidelines:
>
> 1. [positive] Musical Instrument and Inappropriate venue
> - Example: Violin and Football stadium
> - Explanation: Visualize a violin being in a football stadium, creating a striking visual contrast between the delicate instrument and the sports arena.
> 2. [positive] Fruit and Unrelated seasoning
> - Example: Orange and Salt shaker
> - Explanation: Imagine an orange paired with a salt shaker, where the seasoning doesn't match the typical use for the fruit.
> 3. [positive] Spacecraft and Terrestrial landscape
> - Example: Rocket and Desert landscape
> - Explanation: Picture a rocket placed in a desert landscape, highlighting the visual discord between space technology and earthly terrain.
> 4. [positive] Jewelry and Improper storage
> - Example: Diamond necklace and Toolbox
> - Explanation: Visualize a valuable diamond necklace stored in a toolbox, emphasizing the incongruity between luxury jewelry and a utilitarian container.
> 5. [positive] Sports Gear and Wrong sport
> - Example: Baseball bat and Ice hockey rink
> - Explanation: Picture a baseball bat in an ice hockey rink, where the sporting equipment is out of place for the chosen location.
> ......
>
> These patterns create visual contradictions by juxtaposing related concepts in unexpected ways, following the same principles as the given positive patterns.

**Round 5 - Verifying 9 New Patterns**   Due to the lack of guidance from information that has been validated in visual space, the new concept pairs answered by GPT-3.5 don't achieve the expected effect. So human researchers propose a shift toward necessitating concepts within the linguistic domain, rather than demanding concept pairs specifically. In this way, human researchers only need

to select suitable conflicting concepts from two sets of concepts, drawing from their accumulated experience with text-image interactions, and then combine them for verification. This avoids both unguided exploration and excessive guidance:

---

**5**

**Prompt:** What storage is the most unlikely to put jewelry like a pearl ring necklace in? Help me list some of the most unlikely.

**Answer:** When considering containers that are highly unlikely choices for storing a precious item like a pearl ring necklace, we can think creatively. Here are some of the most unlikely containers:

1. Bird's Nest: Placing a delicate pearl ring necklace in a bird's nest would be highly impractical and unexpected.
2. Fish Tank: The underwater environment of a fish tank would be detrimental to a pearl ring necklace.
3. Toilet Bowl: Using a toilet bowl as a storage place for jewelry is not only unhygienic but also absurd.
4. Compost Bin: Pearls and organic waste do not mix, making a compost bin a truly improbable choice.
5. Dog's Food Bowl: Storing jewelry alongside pet food is both unclean and unusual.
6. Fireplace Ashes: The ashes from a fireplace are the last place one would think of storing a delicate necklace.
7. Shoe Sole: Tucking a pearl necklace into the sole of a shoe is an unexpected and risky choice.
8. Fishing Tackle Box: A tackle box designed for fishing equipment is far from suitable for jewelry storage.
9. Hollowed-Out Watermelon: Hiding jewelry inside a watermelon would be a whimsical and impractical choice.
10. Tool Shed Toolbox: Storing jewelry in a toolbox designed for tools is unconventional and risky.

These containers not only provide an answer to your question but also illustrate the unusual and impractical nature of storing a precious item like a pearl ring necklace in them.

---

Human researchers subsequently choose 15 combinations based on their own comprehension. For instance, "Ruby earrings, Bird Nest", "Pearl necklace, Fish tank", "Silver anklet, Toilet bowl", and so on. Then they iterate through the validation process used in Round 2, resulting in $9 \times 15 \times 5 = 675$ text prompts in the form of sentences for Midjourney. Following Midjourney's creation, a total of $675 \times 4 = 2,700$ images are ready to be verified. After thorough verification, we report the number of concept pairs rated as Level 5 for each new pattern in Table 3.

Table 3: New patterns created by GPT-3.5 and the levels of their concept pairs.

| Category | New Pattern | Num of Pairs Rated at Level 5 |
|---|---|---|
| *Category I* | Jewelry and improper storage | 11 / 15 |
| | Flower and unconventional vase | 10 / 15 |
| | Sport and unconventional equipment | 13 / 15 |
| | Musical instrument and inappropriate venue | 11 / 15 |
| | Sports equipment and mismatched locations | 10 / 15 |
| | Household appliances and incorrect room | 8 / 15 |
| *Category II* | Fruits of similar size | 11 / 15 |
| | Sea creatures and land transportation | 4 / 15 |
| | Musical instruments of similar Size | 14 / 15 |

This notably exceeds the proportion of concept pairs directly answered by GPT-3.5 according to patterns without examples.

**Round 6 - Blending Patterns** Two straight lines determine a plane. Once the points on the two lines are defined, it's available to guide GPT-3.5 to expand the new plane:

---

**6**

**Prompt:** I found that the text-to-image model cannot draw some images specified by patterns. For example, the pattern "Beverage and wrong beverage container" ("A and B") and the pattern "Jewelry and Impre storage" ("C and D").

The pattern "Beverage and erroneous container" includes many concept pairs, such as "iced coke in a tea cup", "green tea in a red solo cup", "Orange juice in a coffee mug", and so on.

And another pattern "Jewelry and improper storage" also includes many concept pairs, such as "Diamond necklace in toolbox", "Ruby earings in bird nest", "Silver anklet in toilet bowl", and so on.

I hope to come up with more patterns that cannot be drawn based on these two patterns. Can you help me come up with a few more patterns based on these two patterns. In other words, I mean you to generate new patterns by recombining the items in the two patterns I provided to you, rather than providing new concept pairs in the patterns I gave. You need to use only the two patterns I've provided and regroup things.

In the current case, "A" refers to beverage, "B" refers to beverage containers, "C" refers to Jewelry, and "D" refers to storage. Give me the appropriate 2 patterns "A and D" and "B and C". Where "A" and "D" almost never occur together, and "B" and "C" never occur together. For each new pattern, please give me detailed concept pairs belonging to this pattern that the model may fail to draw such an image.

**Answer:** Certainly, I can help you generate two new patterns "A and D" and "B and C" based on your provided patterns "Jewelry and improper storage" and "Beverage and erroneous container" These new patterns will involve combinations that are less likely to occur together, and the model may have difficulty drawing corresponding images.

Pattern 1: "A and D" - Beverage and Imperial Storage
In this pattern, we combine "Beverage" (A) with "Imperial storage" (D). These two elements are quite different, and the challenge here would be to depict beverages in an imperial storage setting.
1. Champagne in a treasure chest
2. Coffee in a fish tank
3. Soda in a bird nest
4. Tea in an ornate jewelry box
5. Red wine in a toolbox
......

Pattern 2: "B and C" - Beverage Container and Jewelry
This pattern combines "Beverage container" (B) with "Jewelry" (C). These elements typically have no connection, so the challenge would be to depict jewelry inside beverage containers.
1. Diamond ring in a soda can
2. Gold necklace in a coffee thermos
3. Pearl bracelet in a beer mug
4. Sapphire earrings in a wine bottle
5. Ruby brooch in a cocktail shaker
......

These new patterns should provide a creative challenge for the text-to-image model by combining elements that are not usually associated with each other.

---

Human researchers then repeat the verification process used in Round 2 to verify these concept pairs. After thorough verification, we report the number of concept pairs rated as Level 5 for each blended pattern in Table 4. It's quite surprising to discover that the validation results closely align with the expectations of human researchers, indicating that GPT-3.5 successfully integrates two mutually orthogonal patterns.

## B  INTERACTION IN SEQUENTIAL CONCEPT INTRODUCTION

Here, we present the detailed interaction to guide GPT-3.5 to provide the most logical sequence of two concepts and the description of concepts in the Blue Block 7.

Table 4: Blended patterns created by GPT-3.5 and the levels of their concept pairs.

| Category | Blended Pattern | Num of Pairs Rated at Level 5 |
|---|---|---|
| Category I | Jewelry and beverage container | 12 / 15 |
| | Beverage and jewelry storage | 10 / 15 |
| | Flower and beverage container | 10 / 15 |
| | Beverage and flower vase | 11 / 15 |
| | Sport and wrong beverage container | 15 / 15 |
| | Unconventional equipment and beverage | 12 / 15 |
| | Musical instrument and historical sites | 10 / 15 |
| | Endangered species and musical scenes | 9 / 15 |
| | Sports equipment in the wrong type of room | 10 / 15 |
| | Locations with misplaced household appliances | 12 / 15 |
| | Household appliances and location outside of origin | 11 / 15 |
| | Incorrect room and local food | 12 / 15 |
| Category II | Sea creatures and fruits | 9 / 15 |
| | Musical instruments and animals of similar size | 10 / 15 |
| | Land transportations and fruits | 5 / 15 |

---

**7**

**Prompt:** II want you to act as an AI creative assistant. I have two concepts A and B, and a sentence combining these two concepts together. I want you to help me generate a brief visual description of each concept. Namely, a concise description of what a given noun itself looks like, such as color, shape, or other visual features. As short as possible. No background. No other objects.
In addition, if I tend to draw a picture using the sentence, please tell me which concept I should draw first. For example, if a concept is a type of container, we should draw such a container first. If a concept is a type of background, we should draw such background first.
Now, the concept A is "a tea cup", B is "iced coke", and the sentence is "A tea cup filled with iced coke". Please answer me with the descriptions of A and B, and tell me which one to draw first.

**Answer:** Certainly, I can help you.
Concept A: Tea cup - A small, usually white or pastel-colored cup, often with a handle, designed to hold hot beverages.
Concept B: Iced Coke - Dark, fizzy liquid with translucent ice cubes floating within.

To draw the sentence "A tea cup filled with iced coke":
1. Draw the tea cup first (Concept A), as it acts as the container.
2. Then, fill the tea cup with the iced coke (Concept B).

---

## C  SCORE EVALUATION

By convention, evaluation metrics such as Clipscore (Hessel et al., 2021) and Image-Reward (Xu et al., 2023) are usually used to gauge the correspondence between the generations and the input entities quantitatively. These metrics utilize the cosine similarity between embeddings produced by deep neural networks (DNNs). Nonetheless, they often fail to discern numerical values, transparent objects, and other crucial elements readily identifiable by humans (Liu et al., 2022; Theis et al., 2015). In Section 4, our proposed MoCE uses the score difference ($\mathcal{D}$) between the scores of 64 images ($M$) with respect to 2 concepts ($\mathcal{A}$ and $\mathcal{B}$) as the basis for performing a binary search:

$$\mathcal{D} = \mathcal{S}(M, \mathcal{A}) - \mathcal{S}(M, \mathcal{B}) \tag{4}$$

We use this metric to assist in demonstrating the performance of our MoCE. Specifically, we assess the generation performance of both the baseline model (SDXL-1.0) and our MoCE using the metric, $\mathcal{D}$, in Equation 4, where $\mathcal{D}$ is calculated using Clipscore or Image-Reward, called $\mathcal{D}$ - Clipscore[4]

---

[4]To facilitate easier observation by human experts, we demonstrate the established $\mathcal{D}$ - Clipscore at a magnification of $10\times$.

and $\mathcal{D}$ - Image-Reward respectively. Experiments are conducted on both the set of concept pairs at Level 5 and the set of pairs encompassing Levels 1 to 4. We report the experimental results in Table 5. In comparison to the baseline model, $\mathcal{D}$ - Clipscore of images generated by our MoCE is reduced by more than half, and the $\mathcal{D}$ - Image-Reward is reduced by more than $\frac{1}{3}$, in either set of levels. It demonstrates MoCE's ability to effectively restore the lost concepts in images.

Table 5: Score Evaluation for our MoCE using both $\mathcal{D}$ - Clipscore ($\downarrow$) and $\mathcal{D}$ - Image-Reward ($\downarrow$). We use concept pairs originally rated as both Level 5 and Level 1 - 4.

| Original Level | Method | $\mathcal{D}$ - Clipscore $\downarrow$ | $\mathcal{D}$ - Image-Reward $\downarrow$ |
|---|---|---|---|
| Level 1 - 4 | Baseline | 0.91 | 1.10 |
| | MoCE | **0.45** | **0.75** |
| Level 5 | Baseline | 1.09 | 1.21 |
| | MoCE | **0.55** | **0.79** |

## D    RESTORATION VISUALIZATIONS OF LEVEL 5

Here, we demonstrate more visualizations of images of Level 5 restored using our MoCE. In spite of the given additional rich information, Midjourney fails to correctly generate these images. While using the same text prompts, our MoCE successfully retrieves the lost concepts as presented in Figure 5 and 6.

## E    RESTORATION VISUALIZATIONS OF LEVEL 1 - 4

For images restored by baseline models by adding rich information, our MoCE can also easily retrieve the lost concepts and increase the frequency of correct generation, as presented in Figure 7 and 8.

## F    CONCEPT PAIRS SET

Here, we present the concept pairs rated level 5 and 5 sentences describing in our dataset in Table 6. We have also included all the concept pairs and their sentences in each round of Socratic Reasoning in the supplementary material, please refer it for details.

At the same time, we also present their input sequence and the corresponding descriptions provided by GPT-3.5 in Table 7.

Baseline Models | MoCE

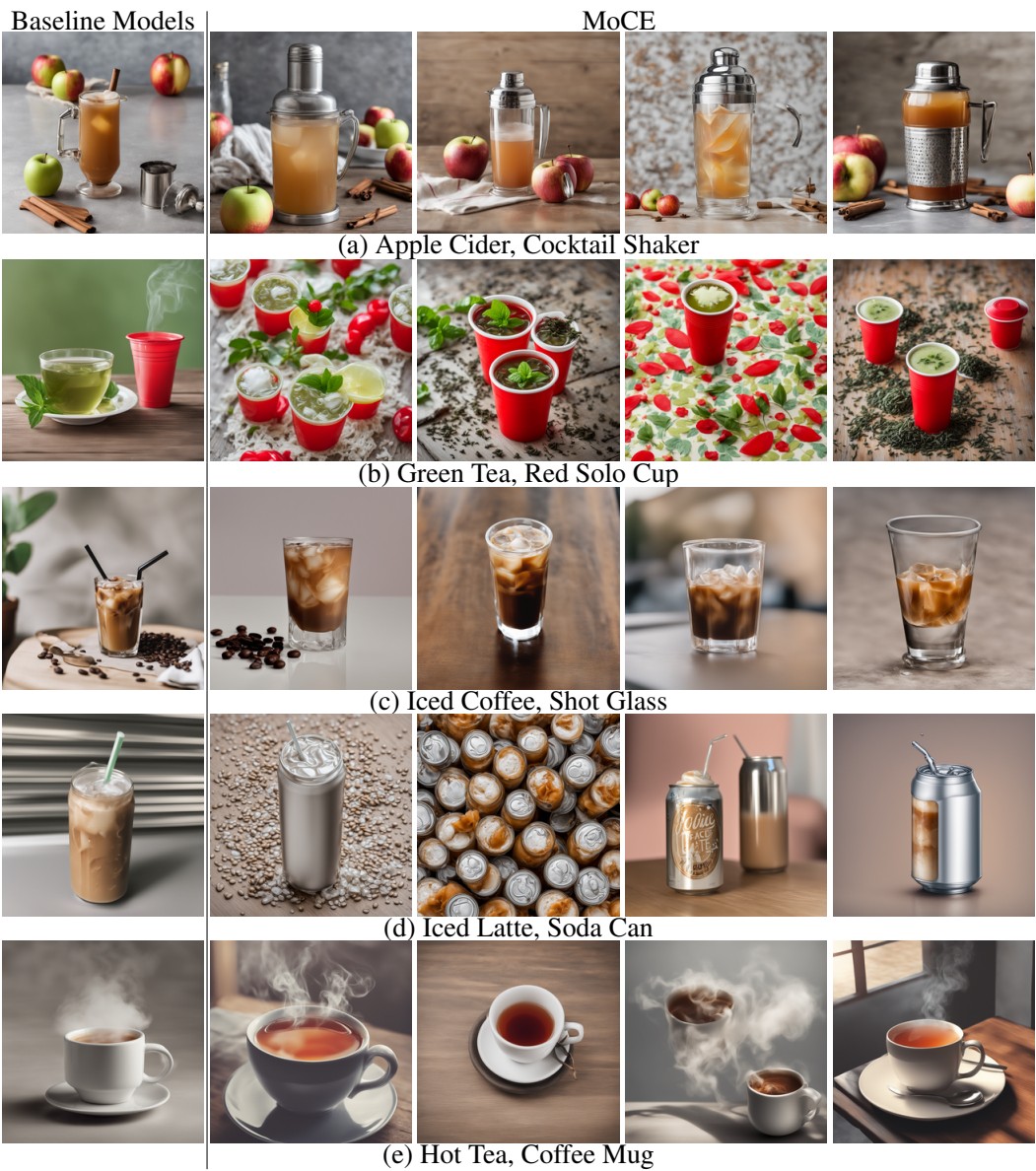

(a) Apple Cider, Cocktail Shaker

(b) Green Tea, Red Solo Cup

(c) Iced Coffee, Shot Glass

(d) Iced Latte, Soda Can

(e) Hot Tea, Coffee Mug

Figure 5: Visualizations of images at Level 5 generated by baseline models and our MoCE.

Baseline Models | MoCE

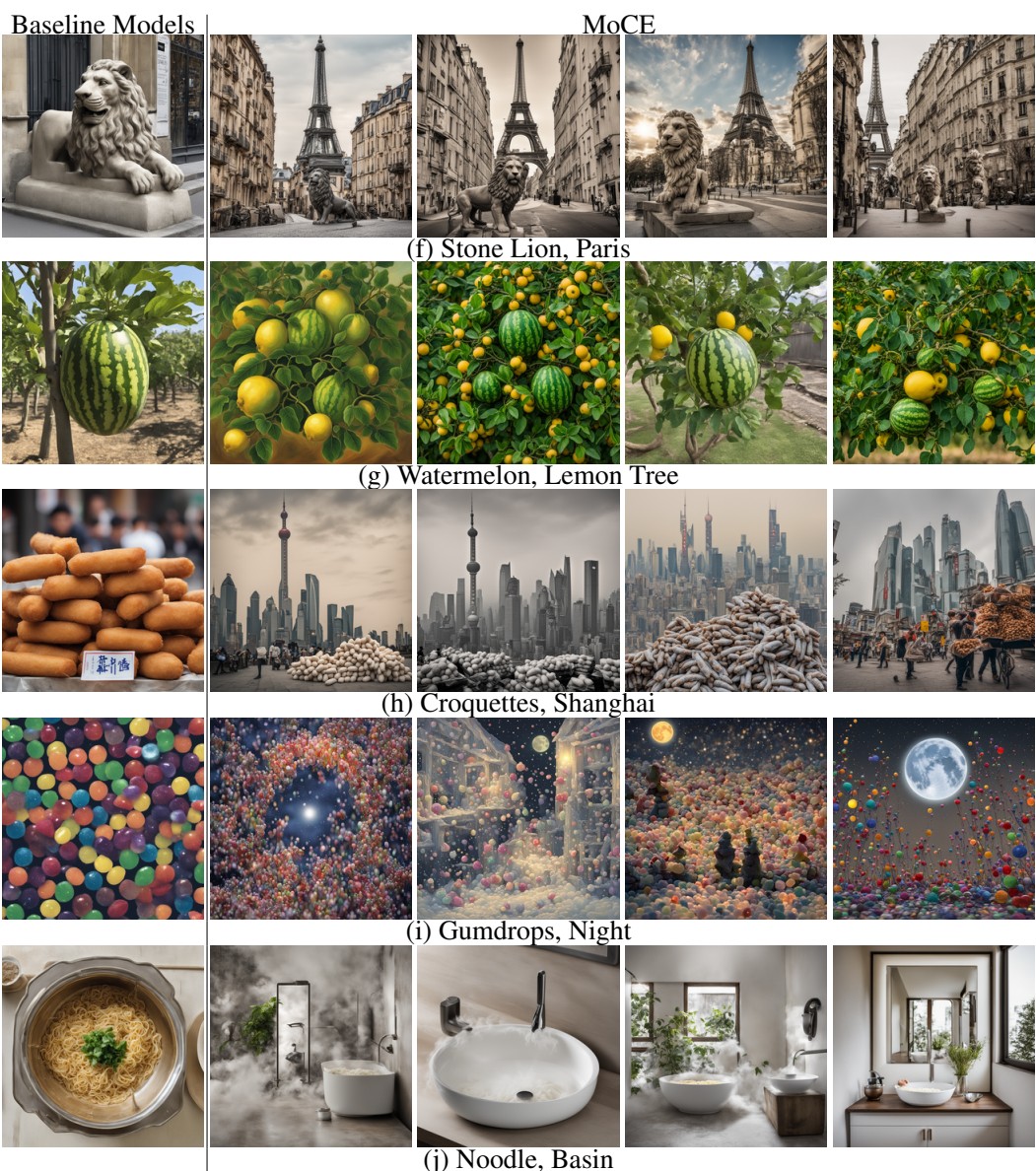

(f) Stone Lion, Paris

(g) Watermelon, Lemon Tree

(h) Croquettes, Shanghai

(i) Gumdrops, Night

(j) Noodle, Basin

Figure 6: Visualizations of images at Level 5 generated by baseline models and our MoCE.

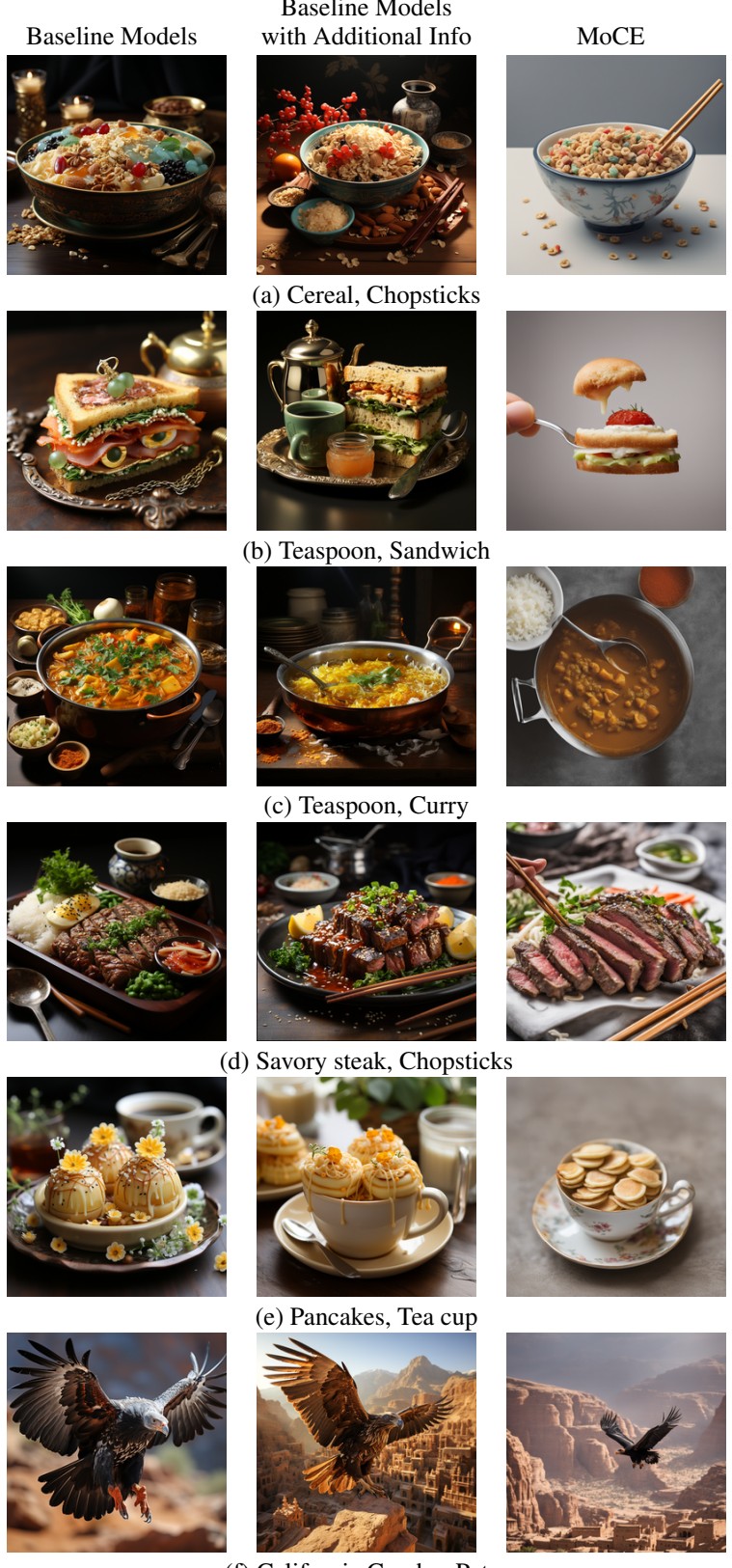

Figure 7: Visualizations of images at Level 1 - 4 generated by baseline models and our MoCE. With additional information, baseline models can indeed generate a small number of correct images. And our MoCE is able to increase the frequency of correct generation.

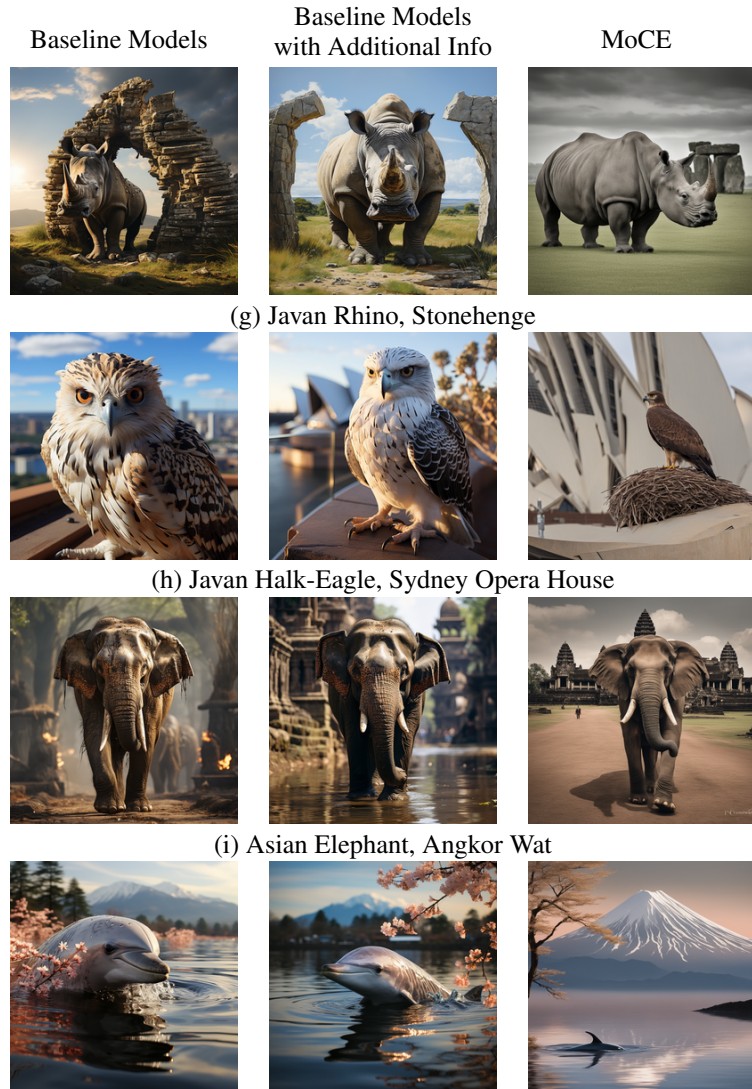

Baseline Models · Baseline Models with Additional Info · MoCE

(g) Javan Rhino, Stonehenge

(h) Javan Halk-Eagle, Sydney Opera House

(i) Asian Elephant, Angkor Wat

(j) Indus River Dolphin, Mount Fuji

Figure 8: Visualizations of images at Level 1 - 4 generated by baseline models and our MoCE. With additional information, baseline models can indeed generate a small number of correct images. And our MoCE is able to increase the frequency of correct generation.

Table 6: Concept pairs rated Level 5 and 5 sentences describing in our dataset.

| Concept $\mathcal{A}$ | Concept $\mathcal{B}$ |
|---|---|
| Bratwurst | Mumbai |

**Prompts**

A Mumbai street vendor grilling Bratwurst sausages.
Bratwurst served on a plate against the backdrop of Mumbai's skyline.
A street in Mumbai lined with vendors selling Bratwurst.
A fusion food cart in Mumbai offering Bratwurst with local flavors.
A bustling Mumbai market with a Bratwurst stall at its heart.

| Concept $\mathcal{A}$ | Concept $\mathcal{B}$ |
|---|---|
| Philippine Crocodile | Neuschwanstein Castle |

**Prompts**

The Neuschwanstein Castle emerging from the murky waters inhabited by Philippine Crocodiles.
A Philippine Crocodile lurking in the moat surrounding the Neuschwanstein Castle.
The Neuschwanstein Castle's reflection distorted by the ripples created by Philippine Crocodiles.
A Philippine Crocodile basking in the shadow of the Neuschwanstein Castle.
The Neuschwanstein Castle nestled amidst the lush habitat of the Philippine Crocodile.

| Concept $\mathcal{A}$ | Concept $\mathcal{B}$ |
|---|---|
| Dolphin | Shaggy fur |

**Prompts**

A dolphin gliding through water with shaggy fur.
Dolphin with shaggy fur surfacing.
A sleek dolphin with a tuft of shaggy fur.
A dolphin's sleek body contrasting with shaggy fur.
Shaggy fur blends with the smooth skin of a dolphin.

| Concept $\mathcal{A}$ | Concept $\mathcal{B}$ |
|---|---|
| White Wine | Shot Glass |

**Prompts**

White wine gracefully filling a shot glass.
A shot glass brimming with white wine.
White wine being poured into a shot glass.
A shot glass holding a generous portion of white wine.
White wine swirling within a crystal-clear shot glass.

| Concept $\mathcal{A}$ | Concept $\mathcal{B}$ |
|---|---|
| Ant | Termite |

**Prompts**

An ant and termite sharing a tiny branch.
An ant marching alongside a termite.
A termite and ant exploring a wooden surface together.
Ant and termite side by side on a wood grain.
An ant and a termite walking together on a piece of wood.

| Concept $\mathcal{A}$ | Concept $\mathcal{B}$ |
|---|---|
| Crop Top | Medieval Armor |

**Prompts**

A medieval knight wearing a crop top under their armor.
The crop top peeking out from beneath the medieval armor.
A knight in medieval armor with a fashionable crop top.
Medieval armor with a stylish crop top underneath.
Crop top worn beneath the heavy medieval armor.

| Concept $\mathcal{A}$ | Concept $\mathcal{B}$ |
|---|---|
| ... | ... |

Table 7: Concept pairs rated Level 5 in our dataset, their descriptions and input sequence provided by GPT-3.5.

| Concept $\mathcal{A}$ | Concept $\mathcal{B}$ |
|---|---|
| Mumbai(first) | Bratwurst |

**Prompt**
A Mumbai street vendor grilling Bratwurst sausages.
**Description of $\mathcal{A}$**
Mumbai is a bustling coastal city in India with a mix of modern skyscrapers,
historical architecture, and a prominent coastline.
**Description of $\mathcal{B}$**
Sausage with a cylindrical shape, typically brown or dark brown in color.

| Concept $\mathcal{A}$ | Concept $\mathcal{B}$ |
|---|---|
| Neuschwanstein Castle(first) | Philippine Crocodile |

**Prompt**
A Philippine Crocodile basking in the shadow of the Neuschwanstein Castle.
**Description of $\mathcal{A}$**
A majestic, fairytale-like palace with tall towers, white walls, and turrets, nestled amidst lush greenery.
**Description of $\mathcal{B}$**
A reptile with scaly, dark green skin and a long, toothy snout.

| Concept $\mathcal{A}$ | Concept $\mathcal{B}$ |
|---|---|
| Shaggy fur(first) | Dolphin |

**Prompt**
The smooth curves of a dolphin contrasted with its shaggy fur.
**Description of $\mathcal{A}$**
Long, unkempt strands of hair.
**Description of $\mathcal{B}$**
Smooth, grayish-blue skin with a streamlined body, a dorsal fin, and a long snout.

| Concept $\mathcal{A}$ | Concept $\mathcal{B}$ |
|---|---|
| Shot Glass(first) | White Wine |

**Prompt**
A shot glass brimming with white wine.
**Description of $\mathcal{A}$**
Small, cylindrical, transparent container.
**Description of $\mathcal{B}$**
Pale yellow liquid in a wine glass.

| Concept $\mathcal{A}$ | Concept $\mathcal{B}$ |
|---|---|
| Ant(first) | Termite |

**Prompt**
An ant and termite sharing a tiny branch.
**Description of $\mathcal{A}$**
Tiny, segmented insect with six legs and often a black or brown color.
**Description of $\mathcal{B}$**
Small, pale, segmented insect.

| Concept $\mathcal{A}$ | Concept $\mathcal{B}$ |
|---|---|
| Medieval Armor(first) | Crop Top |

**Prompt**
A medieval knight wearing a crop top under their armor.
**Description of $\mathcal{A}$**
Metal plates linked together, covering body,
often with a helmet, shield, and sometimes a surcoat displaying heraldry.
**Description of $\mathcal{B}$**
Short, midriff-baring garment.

| Concept $\mathcal{A}$ | Concept $\mathcal{B}$ |
|---|---|
| ... | ... |

