# OpenReview forum: "Lost in Translation: Conceptual Blind Spots in Text-to-Image Diffusion Models"
_ICLR.cc/2024/Conference — Submitted to ICLR 2024_

### Official Review · Reviewer_unnP · 2023-11-01

**Soundness:** 3 good
**Presentation:** 2 fair
**Contribution:** 3 good
**Rating:** 6
**Confidence:** 3

**Summary:**

The paper introduces a novel framework for addressing misalignment errors between visual and textual elements in Text-to-Image (T2I) generation, referred to as 'Conceptual Blind Spots' (CBS). To spot problematic concept pairings, LLMs are employed. Additionally, the paper presents a method called 'Mixture of Concept Experts' (MoCE) to alleviate these identified conceptual blind spots during the diffusion model’s denoising stages. Experimental results illustrate the effectiveness of the proposed framework in mitigating the occurrence of conceptual blind spots.

**Strengths:**

The paper is very interesting and addresses an important limitation of T2I models. Evaluation and results look great. Human evaluation and qualitative analysis further strengthen the proposed framework.

**Weaknesses:**

I do not find any major weaknesses with the proposed work. However, I am curious to know what are the concrete failure cases with the proposed approach. Is there a way to understand the type of prompts the model gets right and the cases the proposed MoCE fails to handle? Another weakness with the paper is the readability. Some of the sections took me multiple readings. Also, in some places, for example in the Human evaluation section, it is not very clear how many test samples are considered. I also find a slight disconnect between the abstract and the three key contributions highlighted in the introduction section. I suggest authors make them more coherent in the final version (for example, there is no mention of dataset contribution in abstract but the dataset is listed as a key contribution in introduction).

**Questions:**

Please see weaknesses.

---

> ### Author Response · Authors · 2023-11-20
> **Response to Reviewer unnP**
>
> Thank you very much for the thoughtful and detailed review. We reply point-by-point here, to begin the discussion. We provide detailed explanations here and will offer a clearer clarification in the updated version of our paper.
>
> 1. Method and Model
>
>     1.1 **We believe that such misalignment issues could be mitigated by more precise image-text pairing during the pre-training process** , similar to the approach taken by DALLE3[2].  We have also supplemented our data with the performance of DALLE3 and the AAE[1] method, which is based on traditional misalignment problems.  The performance of AAE was subpar, while DALLE3 indeed demonstrated superior performance, on par with the SDXL-1.0 model that has been enhanced with MoCE.
>
>     1.2 Our proposed MoCE does indeed encounter some failures. For example, it may overlook the relationship between objects A and B (e.g., when asked for a tea cup of iced coke, it generates a glass of iced coke and an additional teacup), or it merges A and B into a single object (such as generating a panda-themed hairdryer when requested to depict a panda and a hairdryer). We will showcase such examples in the updated version of our paper, possibly in the appendix.
>
> 2. Bad Readability
>
> Thank you very much for pointing out the bad readability of our paper. In the updated version of our paper, we will pay closer attention to this issue, aiming to present our study as clearly as possible. This includes providing clear explanations of key terms and detailing the objectives and processes involved in our dataset collection and so on.
>
> 3. Human Evaluation
> Thank you very much for pointing out your concern. **In human evaluation, we particularly focus on the Level 5 samples within Category I, total of 173. Specifically:**
>
>     3.1 **Category I**: category I represents the misalignment issues involving the hidden object C, as mentioned above. For example, in the case of a teacup (A) and iced coke (B), the frequent association of iced coke (B) with a transparent glass (C) in datasets leads to the replacement of the teacup (A) by the transparent glass (C). While Category II, on the other hand, represents traditional issues involving only A and B, such as requesting an image of a frog and a turtle, the final image contains only two frogs.
>
>     3.2 **Level 5**: GPT first expands each concept pair with five information-rich sentences, and each sentence undergoes four rounds of image generation. This results in a total of 20 images, from which human experts select based on the number of images correctly representing both A and B. A Level 5 rating means all 20 images exhibit misalignment issues, indicating the highest quality samples. All complete instruction prompts are included in the appendix of our paper.
>
>
> | Method | Level 1 | Level 2 | Level 3 | Level 4 | Level 5 | Average Level |
> |-------|-------|-------|-------|-------|-------|-------|
> | SDXL-1.0 Baseline | 0 | 0 | 0 | 0 | 173 | 5 |
> | AAE | 0 | 0 | 6 | 10 | 157 | 4.87 |
> | DALLE3 | 14 | 24 | 31 | 23 | 81 | 3.77 |
> | MoCE (ours) | 11 | 25 | 40 | 71 | 26 | **3.43** |
>
> [1] Hila Chefer, Yuval Alaluf, Yael Vinker, Lior Wolf, and Daniel Cohen-Or. Attend-and-excite: Attention-based semantic guidance for text-to-image diffusion models. ACM Transactions on Graphics (TOG), 42(4):1–10, 2023.
>
> [2] OpenAI. Dall·e 3 system card. 2023.

---

### Official Review · Reviewer_WGri · 2023-11-01

**Soundness:** 2 fair
**Presentation:** 2 fair
**Contribution:** 2 fair
**Rating:** 3
**Confidence:** 4

**Summary:**

In this paper, the authors propose a novel classification for visual-textual misalignment errors called Conceptual Blind Spots. They leverage LLMs and Diffusion Models to detect and correct the CBS.

**Strengths:**

The conceptual blind spot is a relevant and timely problem. Several text-to-image generation models are posied with this limitation. To this extent, the authors build a novel dataset and show its effectiveness.

**Weaknesses:**

The paper is difficult to read. For instance:
1. "In this category, one concept (A or B) demonstrates a dominant relationship, either with an underlying C or" -- it is unclear what A,B, and C represent.
2. How do we come up with the categories and patterns in Table 1
3. "Initially, 259 concept pairs were identified through the collaborative efforts of human researchers, supported by GPT." -- How does GPT help here? What were the guidelines for human researchers?
4. "After rigorous screening, 159 concept pairs attain a Level 5 rating, representing the pinnacle of quality."-- Can use elaborate the screening process?

**Questions:**

See my comments in Weaknesses

---

> ### Author Response · Authors · 2023-11-20
> **Response to Reviewer WGri**
>
> Thank you very much for the thoughtful and detailed review. We reply point-by-point here, to begin the discussion. We provide detailed explanations here and will offer a clearer clarification in the updated version of our paper.
>
> 1. **"Concept Pairs" denotes two main objects A and B in a text segment asked by users. C, while not explicitly indicated, is often bound together with B (or A) in the training dataset.** For example, in the case of a teacup (A) and iced coke (B), the frequent association of iced coke (B) with a transparent glass (C) in datasets leads to the replacement of the teacup (A) by the transparent glass (C). The inclusion of this hidden object C is the main focus of our study.
>
> 2. **Categories and patterns in Table 1 are classified by human experts through meticulous observation.**
>
>     2.1 Specifically, Category I represents the misalignment issues involving the hidden object C, as mentioned above.  Category II, on the other hand, represents traditional issues involving only A and B, such as requesting an image of a frog and a turtle, but the final image contains only two frogs.
>
>     2.2 **Each pattern represents more specific scenarios within each category** , such as "Beverage and Erroneous Container", "Local Cuisine and Non-native Location", ... The existence of patterns can better provide GPT with few-shot guidance.
>
> 3. "What GPT helps, and human researchers guide?"
> **In this process, GPT helps to come up with more misalignment concept pairs, and human researchers provide instruction prompt, few shot and evaluation on images.**
> Specifically, GPT generates new examples that it deems likely to exhibit misalignment issues, based on given misalignment cases. Human researchers are responsible for designing the instruction prompts and providing examples for GPT's few-shot learning. We have included the complete instruction prompt in the appendix of our paper. Furthermore, the examples provided by GPT are sent to Midjourney and SDXL-1.0 for image generation, and human researchers then assess whether misalignment issues occur.
>
> 4. **"Elaborate screening process"**
> Of course, GPT first expands each concept pair with five information-rich sentences, and each sentence undergoes four rounds of image generation. This results in a total of 20 images, from which human experts select based on the number of images correctly representing both A and B. A Level 5 rating means all 20 images exhibit misalignment issues, indicating the highest quality samples. All complete instruction prompts are included in the appendix of our paper.

---

### Official Review · Reviewer_Edr9 · 2023-11-01

**Soundness:** 1 poor
**Presentation:** 1 poor
**Contribution:** 2 fair
**Rating:** 3
**Confidence:** 4

**Summary:**

This paper studies the misalignment between text and image in text-to-image generation. The paper proposes a mixture-of-concept framework, where given a text prompt that contains multiple objects, diffusion models generate objects one by one following a specific order determined by a language model. The paper also collects a dataset of text prompts for evaluation. Experiments show that the proposed method improves faithfulness of generated images compared to the standard stable diffusion baseline.

**Strengths:**

1. The paper collects a valuable benchmark of text prompts for evaluating text-to-image models.
2. The proposed idea of composing generation for each object in a specific order provides insights for future research.

**Weaknesses:**

1. The presentation of the paper is bad. Some important terms such as "conceptual blind spots," "concept pairs," and "Socratic reasoning/questioning" are not clearly defined, which hinders the understanding of the paper. A large portion of the paper is describing the data collection process. However, the overall goal of the dataset is not clearly explained, and the reason behind each round is also not explained. The methodology section is also confusing. How is the proposed metric $D$ used during generation? Why is the binary search algorithm used for generation? Maybe an algorithm table could help explain the method more clearly.
2. The experiment setting is problematic. First, only the standard stable diffusion model is used as baseline. No comparison is provided for related works mentioned in Section 2. Second, the proposed metric $D$ is used during generation as well as for evaluation. So there might be unfair advantages for the proposed method.
3. Some notations are wrong. For example, in Background of Section 4, $I$ shouldn't denote the input image.
4. Several related works are missing, such as [1], [2], [3], [4].

[1] Feng et al., 2022. Training-free structured diffusion guidance for compositional text-to-image synthesis.
[2] Lian et al., 2023. LLM-grounded Diffusion: Enhancing Prompt Understanding of Text-to-Image Diffusion Models with Large Language Models.
[3] Bali et al., 2022. eDiff-I: Text-to-Image Diffusion Models with an Ensemble of Expert Denoisers.
[4] Wu et al., 2023. Harnessing the Spatial-Temporal Attention of Diffusion Models for High-Fidelity Text-to-Image Synthesis.

**Questions:**

Please see weaknesses.

---

> ### Author Response · Authors · 2023-11-20
> **Response to Reviewer Edr9**
>
> Thank you very much for the thoughtful and detailed review. We reply point-by-point here, to begin the discussion. We provide detailed explanations here and will offer a clearer clarification in the updated version of our paper.
> 1. "Missing the clear definition of important terms." Thank you very much for pointing out that some important terms are not clearly defined.
>
>     1.1 **"Concept Pairs" denotes two main objects A and B in a text segment asked by users.** In text-to-image generation, when a text segment contains multiple objects, some may not be accurately represented in the resulting image.  In our study, we focus on scenarios where a text segment contains only two objects.  We refer to these two objects as A and B, collectively termed a "concept pair."
>
>     1.2 **"Conceptual Blind Spots" (CBS) denotes object A (or B) is encroached upon and consequently disappears in the image.** In text-to-image generation, "Conceptual Blind Spots" represents the occurrence of object encroachment. Specifically, this denotes the disappearance of object A (or B), which represents the user's requirement, in the final image. Previous studies primarily focused on scenarios where A is encroached upon by B. However, we have observed that in many instances, A can also be encroached upon by a hidden object C, which often appears bound with B in datasets. For example, in the case of a teacup (A) and iced coke (B), the frequent association of iced coke (B) with a transparent glass (C) in datasets leads to the replacement of the teacup (A) by the transparent glass (C). The inclusion of this hidden object C is the main focus of our study.
>
>     1.3 **"Socratic Reasoning"[1] denotes the process of problem identification, understanding and reasoning using Large Language Models (LLMs), particularly GPT.** We utilize GPT to help us come up with concept pairs that are likely to encounter issues of "Conceptual Blind Spots". The specific procedures for this approach are detailed in Appendix of our paper.
>
> 2. **Missing detailed explanation of the process of data collection** Thank you very much for pointing out that we miss clear explanation for the data collection.
>
>     2.1 **"The overall goal of the dataset" is that we aim to collect CBS samples with inclusion of a hidden object C** mentioned above, a focus not emphasized in previous work. Typically, earlier studies concentrated on scenarios involving only A and B mutually encroaching upon each other.
>
>     2.2 **"The reason behind each round" is that we use a divide-and-conquer strategy, leveraging the extensive knowledge possessed by GPT to expand our dataset.** Specifically, human mind has its limits, hence the need for GPT's assistance. However, GPT often cannot complete the task in one step, leading us to divide the data collection process into several rounds. Initially, we provide CBS examples conceived by human experts and use these as few shot for GPT to generate more examples. Subsequently, human experts categorize these examples into patterns, creating stronger few shot for further rounds. Here, human experts are involved only in the initial two stages. Afterwards, GPT operates like a snowball effect, continually expanding the scope of data collection.
>
> 3. **Missing detailed explanation of our method** Thank you very much for pointing out that we miss clear explanation for the our method. We provide a detailed explanation here, and will give an algorithm table in the updated version of our paper.
>
>     3.1 **A smaller value of D indicates a higher likelihood of objects A and B appearing together in the image.** Specifically, during N-step diffusion sampling, GPT initially suggests the drawing sequence for A and B. Taking the case where A is drawn first, we provide the model with text prompts only for A in the initial x steps and then supply prompts for both A and B in the remaining N-x steps. After the drawing process, we adjust x based on the magnitude and sign of D to find a more optimal x.
>
>     3.2 **We employ a binary search algorithm because we believe that the absolute value of D has a linear relationship with x.** For concept pairs A and B prone to CBS issues, if x is too small, CBS issues will still occur, whereas if x is too large, it limits the time steps available for drawing B. Therefore, we use a binary search algorithm to quickly find the appropriate value of x.
>
> **The 4th, 5th, 6th responses can be found in the next text box.**
>
> [1] Qingxiu Dong, Li Dong, Ke Xu, Guangyan Zhou, Yaru Hao, Zhifang Sui, and Furu Wei. Large language model for science: A study on p vs. np. arXiv preprint arXiv:2309.05689, 2023.

---

> > ### Author Response · Authors · 2023-11-20
> > **Response to Reviewer Edr9**
> >
> > 4. **Problematic experiment setting** Thank you very much for pointing out that the experiment setting is problematic.
> >
> >     4.1 **We have supplemented our research with experimental results from two baseline models, AAE[2] and DALLE3[3]** The results of human evaluations are presented here and will be updated in the paper.
> >
> >     4.2 **We greatly appreciate your feedback regarding the evaluation metrics.** Given the tendency of automatic scoring to be ineffective in addressing CBS issues, we will focus on human evaluation, encompassing both the experiments already included in the paper and the additional ones we have supplemented.
> >
> > 5. Thank you very much for pointing out that some notations are wrong. We will make modifications and improvements in the updated version of our paper.
> >
> > 6. Thank you very much for pointing out that several related works are missing. We will make modifications and improvements in the updated version of our paper.
> >
> >
> > | Method | Level 1 | Level 2 | Level 3 | Level 4 | Level 5 | Average Level |
> > |-------|-------|-------|-------|-------|-------|-------|
> > | SDXL-1.0 Baseline | 0 | 0 | 0 | 0 | 173 | 5 |
> > | AAE | 0 | 0 | 6 | 10 | 157 | 4.87 |
> > | DALLE3 | 14 | 24 | 31 | 23 | 81 | 3.77 |
> > | MoCE (ours) | 11 | 25 | 40 | 71 | 26 | **3.43** |
> >
> > [2] Hila Chefer, Yuval Alaluf, Yael Vinker, Lior Wolf, and Daniel Cohen-Or. Attend-and-excite: Attention-based semantic guidance for text-to-image diffusion models. ACM Transactions on Graphics (TOG), 42(4):1–10, 2023.
> >
> > [3] OpenAI. Dall·e 3 system card. 2023.

---

### Meta-Review · Area_Chair_RenX · 2023-12-14

**Metareview:**

This paper introduces a promising concept in text-to-image generation with its mixture-of-concept framework and the collection of a novel dataset for evaluating text-to-image models. These aspects are commendable, offering fresh insights and tools for future research in the field. However, despite these strengths, the paper falls short in key areas, leading to a decision to reject. Issues with clarity in presentation, particularly the undefined critical terms, hinder understanding of its contributions. Methodological explanations are insufficient, and the experimental setup lacks a comprehensive comparison with related works. These significant shortcomings, particularly in presentation and methodological rigor, overshadow the paper's positive aspects and prevent it from meeting the conference's acceptance criteria.

**Justification For Why Not Higher Score:**

Same as above.

**Justification For Why Not Lower Score:**

N/A

---

### Decision · Program_Chairs · 2024-01-16

Reject